# Topological data analysis of human brain networks through order statistics

**Soumya Das**[ID]**, D. Vijay Anand, Moo K. Chung**[ID]*

Department of Biostatistics and Medical Informatics, University of Wisconsin-Madison, Madison, WI, United States of America

* mkchung@wisc.edu

## Abstract

Understanding the common topological characteristics of the human brain network across a population is central to understanding brain functions. The abstraction of human connectome as a graph has been pivotal in gaining insights on the topological properties of the brain network. The development of group-level statistical inference procedures in brain graphs while accounting for the heterogeneity and randomness still remains a difficult task. In this study, we develop a robust statistical framework based on persistent homology using the *order statistics* for analyzing brain networks. The use of order statistics greatly simplifies the computation of the persistent barcodes. We validate the proposed methods using comprehensive simulation studies and subsequently apply to the resting-state functional magnetic resonance images. We found a statistically significant *topological* difference between the male and female brain networks.

**Data Availability Statement:** The data is available from the public domain HCP database: http://www.humanconnectomeproject.org The code is available at https://github.com/laplcebeltrami/orderstat.

## Introduction

Modeling the human brain connectome as graphs has become the cornerstone of neuroscience, enabling an efficient abstraction of the brain regions and their interactions [1, 2]. Graphs offer the simplistic construct with only a set of nodes and edges to describe the connectivity of the brain network [3]. The generalizability of graph representation allows one to obtain quantitative measures across multiple spatio-temporal scales ranging from the node level up to the whole network level [4, 5]. To build the graph representation of brain networks, the whole brain is usually parcellated into hundreds of disjoint regions, which serves as nodes and the edges are associated with weights that indicate the strength of connection between the brain regions [6]. The graph theory based models provide reliable measures such as small-worldness, modularity, centrality and hubs [7–9]. However, these measures are often affected by the choice of arbitrary thresholds on the edge weights and thus make comparisons across networks difficult [10, 11]. To overcome this issue, the topological data analysis (TDA) has emerged to be a powerful method to systematically extract information from hierarchical layers of abstraction [12–15].

Persistent homology (PH), one of the TDA techniques, provides a coherent framework for obtaining topological invariants or features such as connected components and cycles at

**Funding:** For this study, we received funding from National Institute of Health (NIH) R01 EB02875 and National Science Foundation (NSF) MDS-2010778. All the authors (Das, Anand, Chung) received salaries from these two grants. However, the design of the study, the analysis of data and writing of papers are all done by the authors independent of the funders.

**Competing interests:** The authors have declared that no competing interests exist.

different spatial resolutions [16–19]. These topological invariants are often used to provide robust quantification measures to assess the topological similarity between networks [6]. Mostly the persistent barcodes are represented as persistent landscapes or diagrams and their distributions are used to compute a topological distance measure [20]. The PH based topological distances are found to consistently outperform traditional graph based metrics [21]. The main idea of using PH to brain networks is to generate a sequence of nested networks over every possible threshold through a graph filtration, which builds the hierarchical structure of the brain networks at multiple scales [10, 22–24].

In the graph filtration, a series of nested graphs containing topological information at different scales are produced. During the graph filtration, some topological features may live longer, whereas others *die* quickly. The filtration process tracks the birth, death and persistence of the topological features. The lifespans or *persistence* of these features are directly related to the topological properties of networks. The collection of intervals from births to deaths that defines persistences are called the barcodes which characterizes the topology of an underlying dataset [14]. The persistent diagram displays the paired births and deaths as scatter points [16, 20, 25–29]. The Betti curves, which counts the number of such features over filtrations, provide comprehensive visualizations of these intervals [6]. Thus, it is instructive to develop a statistical inference procedure using the persistent barcodes in order to compare across different groups and achieve meaningful inferences. This requires the statistical version of TDA [27, 30]. [30] worked on computing confidence bands through bootstraps. [27] introduced persistent landscapes which lies in a vector space, where the sample mean and variance can be computed and thus enable a proper statistical inference. [31] worked on the hypothesis testing on the Gaussian kernel smoothing on persistent diagrams Analogous to the persistent barcodes, we also have their stochastic versions referred as the expected persistent barcodes. However, computing them requires complex theoretical constructs and they are generally approximated [32–34].

Since the real brain networks are often affected by heterogeneity and intrinsic randomness [35, 36], it is challenging to build a coherent statistical framework to transform these topological features as quantitative measures to compare across different brain networks by averaging or matching [37]. The brain networks are inherently noisy which makes it even harder to establish similarity across networks. Thus, there is a need to develop a statistical model that accounts for the randomness and provides consistent results across networks. The statistical models based on the distributions are expected to be more robust and less affected by the presence of outliers. To this end, we use the concept of random graph to analyze brain networks across a population.

The random graphs have been investigated by many authors [38–42]. A graph whose features related to nodes and edges are determined in a random fashion is called a random graph. The theory of random graphs lies at the intersection between graph theory and probability theory. They are usually described using a probability distribution or a stochastic process that generates them [43, 44]. The homology of random graphs have been studied studied by Kahle in particular. [45] investigated the connectivity of neighborhood complex of a random graph. [46] studied the expected topological properties of Rips complexes built on randomly distributed points in $\mathbb{R}^d$. [47] worked on the central limit theorem for Betti numbers of random simplical complexes. Random graphs are often encountered in graphical models, which build probabilistic models on the conditional dependency structures between nodes [48, 49]. However, topology is rarely investigated in graphical models.

In this paper, we propose a more adaptable random graph model for brain networks. We consider a random complete graph, where all the nodes are connected with its edge weights

randomly drawn from a continuous distribution. The consideration of a complete graph model simplifies building graph filtration straightforward [22, 37]. We then compute the expected 0D and 1D barcodes through the order statistics [50–55]. The use of order statistics in computing persistent homology features such as persistent barcodes and Betti numbers can drastically speed up the computation. Further, we propose the *expected topological loss* (ETL), which quantifies the 0D and 1D barcodes obtained through order statistics. We use the ETL as a test statistic to determine the topological similarity and dissimilarity between networks. The proposed random graph model and corresponding ETL methods are validated using extensive simulation studies with the ground truths. Subsequently, the method is applied to the resting-state functional magnetic resonance images (rs-fMRI) of the human brain.

## Materials and methods

### Data

We considered a resting-state fMRI dataset collected as part of the Human Connectome Project (HCP) [56, 57]. The dataset consisted of fMRI scans of 400 subjects (168 males and 232 females) over approximately 14.5 minutes using a gradient-echoplanar imaging sequence with 1200 time points [24, 37]. Informed consent was obtained from all participants by the Washington University in St. Louis institutional review board [58]. The ethics approval for using the HCP data was obtained from the local ethics committee of University of Wisconsin-Madison.

The human brain can be viewed as a weighted network with its neurons as nodes. However, considering a high number of neurons ($\sim 10^{12}$) in a human brain, the traditional brain imaging studies parcellate the brain into a manageable number of mutually exclusive regions [59–61]. These regions are then considered as nodes while the strength of connectivity between these regions are edges. For the considered dataset, the Automated Anatomical Labeling (AAL) template was employed to parcellate the brain volume into 116 non-overlapping anatomical regions [62] and the fMRI across voxels within each brain parcellation were averaged. This resulted 116 average fMRI time series with 1200 time points for each subject. Further, we removed fMRI volumes with significant head movements [63] because such movements are shown to produce spatial artifacts in functional connectivity [64–66].

### Simplicial complex

A *simplex* is a generalization of the notion of a triangle or tetrahedron to arbitrary dimensions. A 0-simplex is a point, a 1-simplex is a line segment, and a 2-simplex is a triangle. In general, a $k$-simplex $S_k$ is a convex hull of $k + 1$ affinely independent points $u_0, u_1, \ldots, u_k \in \mathbb{R}^k$:

$$S_k = \left\{ \theta_0 u_0 + \cdots + \theta_k u_k \ \middle| \ \sum_{i=0}^{k} \theta_i = 1, \theta_i \geq 0 \ \text{for} \ i = 0, \cdots, k \right\}.$$

Whereas, a simplicial complex $\mathcal{K}$ is a set of simplices that satisfies the following two conditions. (1) Every face of a simplex from $\mathcal{K}$ is also in $\mathcal{K}$. (2) The non-empty intersection of any two simplices $S_1, S_2 \in \mathcal{K}$ is a shared face [67]. We call a simplicial complex consisting of up to $k$-simplices a $k$-skeleton. Since graphs are a collection of nodes (0-simplices) and edges (1-simplices), they are 1-skeleton simplicial complexes. In a 1-skeleton, 0-dimensional (0D) holes are *connected components* while the 1-dimensional (1D) holes are *cycles*. A cycle or loop in a graph is a path that starts and ends at the same node but no other nodes in the path are overlapping. There is no higher dimensional homology beyond dimensions 0 and 1 in 1-skeleton [37].

## Graph filtration

The brain networks are traditionally represented and analyzed as a graph, a 1-skeleton consisting of only nodes and edges [59–62, 68]. The main focus of functional brain network analysis is quantifying and modeling the pairwise interaction between brain regions, which is usually called the *effective connectivity* [69–72]. Thus, we limited our algebraic representation of brain networks to graphs. Compared to the vast body of studies analyzing brain networks as graphs, modeling them as higher order simplicial complexes are only few [17, 73–75]. We used the graph filtration, which iteratively builds nested subgraphs of the original graph in a hierarchical manner [22]. Currently, this is the most often used filtration in analyzing brain networks due to its simplicity.

Consider a weighted graph $G(p, \boldsymbol{w})$, where $p$ is the number of nodes and $\boldsymbol{w} = (w_1, \ldots, w_q)^\top$ is a $q$-dimensional vector of edge weights with $q = (p^2 - p)/2$. The binary graph $G_\epsilon(p, \boldsymbol{w}_\epsilon)$ of $G(p, \boldsymbol{w})$ has binary edge weight $w_{\epsilon,i}$:

$$w_{\epsilon,i} = \begin{cases} 1, & \text{if } w_i > \epsilon, \\ 0, & \text{otherwise.} \end{cases}$$

The binary network $G_\epsilon(p, \boldsymbol{w}_\epsilon)$ is a 1-skeleton. In 1-skeleton, 0-dimensional (0D) holes are *connected components* while the 1-dimensional (1D) holes are *cycles*. There is no higher dimensional homology beyond dimensions 0 and 1. The number of connected components and the number of independent cycles in a graph are referred to as the 0th Betti number ($\beta_0$) and 1st Betti number ($\beta_1$) respectively. For 1-skeletons, there is an efficient 1D filtration method called the *graph filtration*, which filters at the edge weights from $-\infty$ to $\infty$ in a sequentially increasing manner [6, 37]. The graph filtration of $G$ is defined as a collection of nested binary networks

$$G_{\epsilon_0} \supset G_{\epsilon_1} \supset \cdots \supset G_{\epsilon_k}$$

over increasing filtration values $\epsilon_0 < \epsilon_1 < \cdots < \epsilon_k$. We used the edge weights as the filtration values to make the graph filtration unique [6].

During the graph filtration, edges are deleted one at a time from the lowest edge weight to the highest. The deletion of an edge disconnect the graph into at most two. Thus, the number of connected components ($\beta_0$) stays the same or increases at most by one. Euler characteristic $\chi$ of the graph is given by [76]

$$\chi = \beta_0 - \beta_1 = \# \text{ of nodes} - \# \text{ of edges.}$$

Thus the change of Euler characteristic $\Delta\chi$ over the filtration is given by

$$\Delta\chi = \Delta\beta_0 - \Delta\beta_1 = 1,$$

where the change of $\beta_0$ is $\Delta\beta_0 = 0$ or 1. Subsequently the change of $\beta_0$ is $\Delta\beta_1 = -1$ or 0. The number of cycles decrease at most by 1 [6, 77].

## Birth-death decomposition

When we increase the filtration value $\epsilon$, either one new connected component appears or one cycle disappears [6]. Once a connected component is born, it never dies implying an infinite death value. On the other hand, all the cycles are considered to be born at $-\infty$. Therefore, we simply ignore the infinite death values of the connected components and the negative infinite birth values of the cycles and build the computation framework based on only the birth (death) values of the connected components (cycles) [37]. Also, the number of connected components (or cycles) is non-decreasing (or non-increasing) as $\epsilon$ increases. Subsequently, the 0D

barcode is given by a set of increasing birth values:

$$B(G) : \epsilon_{b_1} < \cdots < \epsilon_{b_{m_0}},$$

and the 1D barcode is given by a set of increasing death values:

$$D(G) : \epsilon_{d_1} < \cdots < \epsilon_{d_{m_1}}.$$

By tracing the birth values of connected components and the death values of cycles *together*, we can characterize the topology of the graph.

The above 0D and 1D barcodes summarize the persistences of connected components and cycles and are often visualized using persistent diagrams [16, 25–27] and Betti curves. The Betti curves plot the Betti numbers with respect to the filtration values. Since the Betti numbers are monotonic, the Betti curve is a step function with a one-unit jump (or drop) at every birth (or death) values. The total number of finite birth values of connected components and the total number of death values of cycles are

$$m_0 = p - 1 \quad \text{and} \quad m_1 = \frac{(p-1)(p-2)}{2},$$

respectively [37]. The number of connected components ($\beta_0$) and cycles ($\beta_1$) in the complete graph $G_{-\infty}$ are 1 and $m_1$ respectively. We note that every edge weight must be in either 0D barcode or 1D barcode as summarized in the following theorem [37].

**Theorem 1 (*Birth-death decomposition*).** *The set of 0D birth values $B(G)$ and 1D death values $D(G)$ partition the edge weight vector $\boldsymbol{w}$ such that $B(G) \cup D(G) = \boldsymbol{w}$ and $B(G) \cap D(G) = \phi$. The cardinalities of $B(G)$ and $D(G)$ are $p - 1$ and $(p - 1)(p - 2)/2$, respectively.*

The schematic of graph filtration and birth-death decomposition for a random graph is presented in Fig 1. The cycles we identify using the birth-death decomposition algebraically independent of each other and hence form a basis for cycles [24, 37]. In binary graph $G_{W_3}$ in Fig 1, there is one cycle consisting of edge weights $W_{(4)}$, $W_{(5)}$ and $W_{(6)}$. The cycle can be algebraically represented as $[W_{(4)}] + [W_{(5)}] + [W_{(6)}]$ with the convention of putting clockwise orientation along the edges. In the complete graph $G_{-\infty}$, there are three independent cycles

$$
\begin{aligned}
C_1 &= [W_{(4)}] + [W_{(5)}] + [W_{(6)}] \\
C_2 &= -[W_{(5)}] + [W_{(3)}] + [W_{(2)}] \\
C_3 &= [W_1] + [W_6] + [W_3].
\end{aligned}
$$

All other cycles can be represented as a linear combination of $C_1$, $C_2$ and $C_2$. For instance,

$$
\begin{aligned}
[W_{(4)}] + [W_{(3)}] + [W_{(2)}] + [W_{(6)}] &= C_1 + C_2 \\
-[W_{(1)}] + [W_{(2)}] + [W_{(4)}] &= -C_3 + C_1 + C_2.
\end{aligned}
$$

The total number of algebraically independent cycles is the 1st Betti number $\beta_1$, which is equivalent to the number of death values of cycles. For a complete graph with $p$ nodes, the total number of edges is $p(p - 1)/2$. In a graph filtration, the total number of birth values of connected components equals to the number of edges $p - 1$ in the maximum spanning tree. From the birth-death decomposition, the remaining edges contribute to the death values of

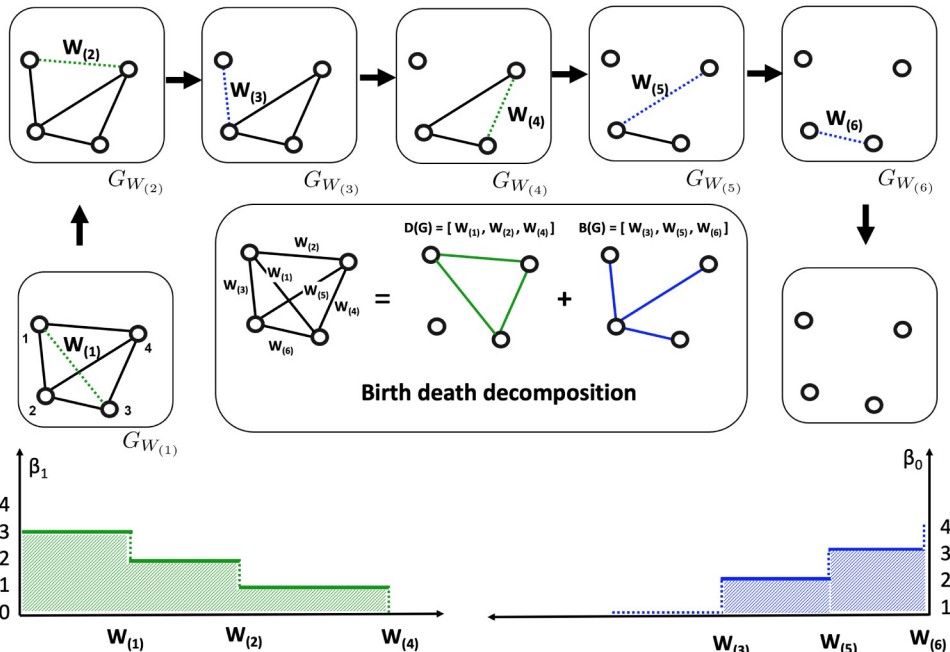

**Fig 1. Schematic of graph filtration and persistent barcodes computation.** We consider a random weighted graph with $p = 4$ nodes, where the number of edges is $q = p(p - 1)/2 = 6$. The random edge weights are $\{W_1, W_2, \ldots, W_6\}$. We order them using the order statistic as $W_{(1)} < W_{(2)} < \cdots < W_{(6)}$. We remove each edge of the random graph one at a time in the graph filtration and construct the random birth and death sets of the connected components and cycles, respectively. The Betti-0 (lower right) and Betti-1 (lower left) curves are drawn using the birth and death sets. The blue and green shaded areas represent the areas under Betti-0 and Betti-1 curves. Later, we will consider the area under Betti-0 curve to quantify the curve and construct a test statistic to discriminate between two groups of networks.

cycles. The remaining number of edges are [37]

$$m_1 = \frac{p(p - 1)}{2} - (p - 1) = \frac{(p - 1)(p - 2)}{2}.$$

The connected components characterize the modular structure or shape of a network whereas the cycles are loops in a network [24, 78]. In [24], the authors focused specifically on cycles in a brain network as they embed higher order signal transmission paths to provide insights of the functioning of the brain. The presence of more cycles in a network indicates a dense connection with stronger connectivity. The cycles in the brain network not only determines the propagation of information but also controls the feedback [79]. The connected components and cycles provide dependent but complementary information about the network.

## Wasserstein distance on barcodes

Since there is no higher dimensional homology beyond dimensions 0 and 1 in 1-skeleton, the 0D and 1D barcodes together can characterize the topology of a network [14]. Therefore, the topological similarity between two such networks can be quantified using a distance metric between the corresponding 0D or 1D barcodes [80]. Often used metric is the Wasserstein distance [23, 81–83]. Let $G_1(p, \boldsymbol{u})$ and $G_2(p, \mathbf{v})$ be two networks and the corresponding barcodes

(or persistent diagrams) be $P_1$ and $P_2$. Then, the 2-*Wasserstein distance* on barcodes is given by

$$\mathcal{D}(P_1, P_2) = \inf_{\tau: P_1 \to P_2} \left( \sum_{x \in P_1} \|x - \tau(x)\|^2 \right)^{1/2}$$

over every possible bijection $\tau$ between $P_1$ and $P_2$ [23, 84, 85]. For graph filtrations, barcodes are 1D scatter points. Therefore, the bijection $\tau$ can be simplified to the $\mathbb{L}_2$ norm between the sorted birth values of connected components or the sorted death values of cycles [23].

**Theorem 2** *Let $G_1$ and $G_2$ be two networks with p nodes and*

$$\{\epsilon_{b_1}^{(k)} < \cdots < \epsilon_{b_{m_0}}^{(k)}\} \quad \text{and} \quad \{\epsilon_{d_1}^{(k)} < \cdots < \epsilon_{d_{m_1}}^{(k)}\}$$

*be the birth and death sets of the network $G_k$, k = 1, 2. Then, the 2-Wasserstein distance between the 0D barcodes for graph filtration is given by*

$$\mathcal{D}_0^2(P_1, P_2) = \left[ \sum_{i=1}^{m_0} \left( \epsilon_{b_i}^{(1)} - \epsilon_{b_i}^{(2)} \right)^2 \right]^{1/2},$$

*and the 2-Wasserstein distance between the 1D barcodes is*

$$\mathcal{D}_1^2(P_1, P_2) = \left[ \sum_{i=1}^{m_1} \left( \epsilon_{d_i}^{(1)} - \epsilon_{d_i}^{(2)} \right)^2 \right]^{1/2}.$$

## Expected persistent barcodes of random graph

Consider random graph $\mathcal{G}(p, \boldsymbol{W})$, where its edge weights are drawn independent and identically from a distribution function $F_W$. $p$ is the number of nodes and $\boldsymbol{W} = (W_1, \ldots, W_q)^\top$ is a $q$ dimensional vector of random weights with $q = (p^2 - p)/2$. The considered graph is complete and its edge weights are drawn randomly from a continuous distribution. To be mathematically precise, the considered random graph is *almost surely* complete. Since we the edge weights are drawn from a continuous distribution, the probability of having zero edge weight is *nil*. Fig 2 displays weighted brain networks randomly drawn from Beta distributions.

If we apply a graph filtration on the random weighted graph $\mathcal{G}(p, \boldsymbol{W})$, we have a set of random birth values of connected components (or random 0D barcode) and a set of random death values of cycles (or random 1D barcode). Since the notions of random birth and death values are abstract, it is important to turn them into deterministic topological descriptors. As often, one of the simplest way to turn a random object into a deterministic summary is to consider its average behavior. To that end, we study the expected birth and death values (or expected persistent barcodes) as follows.

Let $\mathcal{G}(p, \boldsymbol{W})$ be a random graph and its sorted *random* edge weights be

$$W_{(1)} < W_{(2)} < \cdots < W_{(q)},$$

where the subscript $(i)$ indicates the $i$th smallest edge weight. For instance, $W_{(1)} = \min_{1 \le i \le q} W_i$ is the smallest edge weight while $W_{(q)} = \max_{1 \le i \le q} W_i$. is the largest edge weight. Order statistics can be formulated by modeling indices $(i)$ using random permutations while the actual edge weights are fixed nonrandom quantity. However, in order statistics, the indices $(i)$ themselves are not considered as random but fixed [86, 87]. They simply indicates the order the random

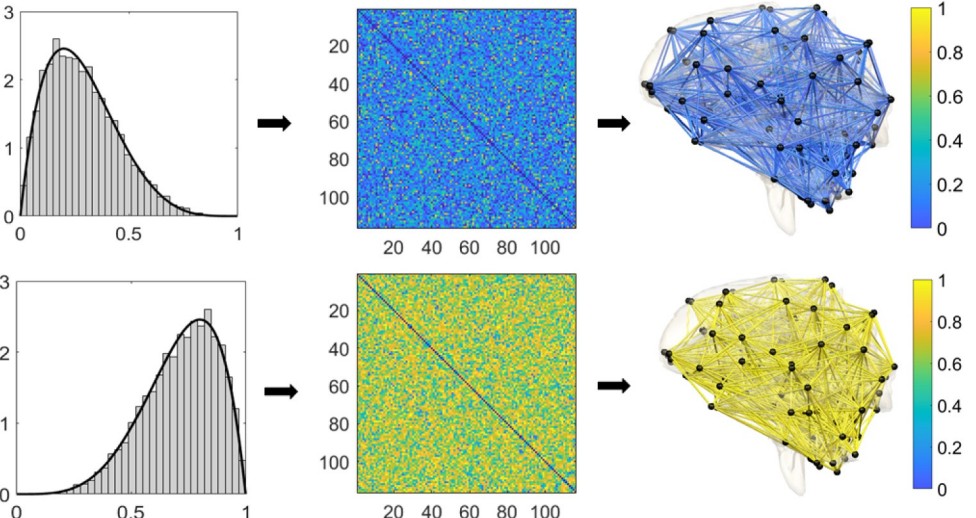

**Fig 2. Visualization of simulated brain networks with 116 nodes.** Left: The empirical density functions of simulated edge weights from Beta(2, 5) (top) and Beta(5, 2) (bottom) distributions. Middle: The 116 × 116 correlation matrices constructed using the simulated edge weights. Right: Human brain networks with the simulated edge weights. Since correlation networks are too dense for visualization, we only displayed edges with values below 0.1 and above 0.9.

variables are indexed. Only what is in the $i$th variable is considered as random. In this study, we will follow the traditional convention in order statistics.

Let the *random* birth and death values of the connected components and cycles be

$$B(\mathcal{G}) : W_{(i_1)} < W_{(i_2)} < \cdots < W_{(i_{m_0})} \quad \text{and} \quad D(\mathcal{G}) : W_{(j_1)} < W_{(j_2)} < \cdots < W_{(j_{m_1})},$$

where $m_0 = p - 1$ and $m_1 = (p - 1)(p - 2)/2$. Then, the expected birth and death values are given by

$$B(\mathcal{G}) : \mathbb{E}(W_{(i_1)}) < \mathbb{E}(W_{(i_2)}) < \cdots < \mathbb{E}(W_{(i_{m_0})})$$

and

$$D(\mathcal{G}) : \mathbb{E}(W_{(j_1)}) < \mathbb{E}(W_{(j_2)}) < \cdots < \mathbb{E}(W_{(j_{m_1})}),$$

where $\mathbb{E}$ indicates the standard *expectation operator* on a *random* weight.

In order to compute the expected birth and death values, we provide an explicit expression for $\mathbb{E}(W_{(k)})$, for $k = 1, \ldots, q$, through Theorem 3 below.

**Theorem 3** *Let the edge weights* $\boldsymbol{W} = \{W_1, W_2, \ldots, W_q\}$ *of a random graph* $\mathcal{G}(p, \boldsymbol{W})$ *be drawn from a distribution with cumulative distribution function (cdf)* $F_W$ *and probability density function (pdf)* $f_W$. *Then, the expectation of the kth edge wight* $W_{(k)}$ *can be approximated by*

$$\mathbb{E}\left(W_{(k)}\right) \approx F_W^{-1}\left(\frac{k}{q+1}\right), \quad k = 1, \ldots, q.$$

**Proof** Since the edge weights $W_1, W_2, \ldots, W_q$ are drawn from a distribution with a cdf $F_W$ and a pdf $f_W$, the pdf of the $k$th order statistic $W_{(k)}$ is given by

$$W_{(k)} \sim \frac{q!}{(k-1)!(q-k)!} f_W(x)\{F_W(x)\}^{k-1}\{1 - F_W(x)\}^{q-k}. \tag{1}$$

$W_{(k)}$ does not follow a well-known distribution and, therefore, the computation of its mean and variance is difficult. However, [88] showed that the $r$th sample quantile of $\{W_1, W_2, \ldots, W_q\}$ is asymptotically normally distributed:

$$W_{([(q+1)r])} \sim \mathcal{AN}\left(F_W^{-1}(r), \frac{r(1-r)}{(q+1)[f_W\{F_W^{-1}(r)\}]^2}\right)$$  (2)

for large $q$, where $\mathcal{AN}$ stands for *asymptotic normal* distribution. Thus, the approximate mean and variance of $W_{(k)}$ can be found from (2) by letting $r = k/(q+1)$:

$$\mathbb{E}\left(W_{(k)}\right) \approx F_W^{-1}\left(\frac{k}{q+1}\right) \quad \text{and} \quad \text{var}\left(W_{(k)}\right) \approx \frac{\frac{k}{q+1}\left(1 - \frac{k}{q+1}\right)}{(q+1)\left[f_W\left\{F_W^{-1}\left(\frac{k}{q+1}\right)\right\}\right]^2}.$$

The variance will be later used in computing confidence intervals.

Now, we use Theorem 3 and provide expressions for the expected birth and death values in Theorem 4 below.

**Theorem 4** *Let $\mathcal{G}(p, W)$ be a random graph, where its edge weights are drawn from a cdf $F_W$. Then, the expected birth values of the connected components of $\mathcal{G}(p, W)$ are given by*

$$F_W^{-1}\left(\frac{i_1}{q+1}\right) < \cdots < F_W^{-1}\left(\frac{i_{m_0}}{q+1}\right)$$

*and the expected death values of the cycles of $\mathcal{G}(p, W)$ are given by*

$$F_W^{-1}\left(\frac{j_1}{q+1}\right) < \cdots < F_W^{-1}\left(\frac{j_{m_1}}{q+1}\right).$$

The proof follows Theorem 3. As a special case of Theorem 4, we show that if the edge weights follow uniform distribution in [0, 1], the expected birth and death values have a more simplified and an exact form. The expected birth values of the connected components are given by

$$\frac{i_1}{q+1} < \cdots < \frac{i_{m_0}}{q+1}$$

and the expected death values of the cycles are given by

$$\frac{j_1}{q+1} < \cdots < \frac{j_{m_1}}{q+1}.$$

Then the distribution of the $k$th order statistic simplifies to

$$W_{(k)} \sim \frac{q!}{(k-1)!(q-k)!} w^{k-1}(1-w)^{q-k},$$

which is the distribution of the Beta distribution with parameters $k$ and $q + 1 - k$. Since the mean of a Beta$(k, q + 1 - k)$ distribution has an exact form of $k/(q+1)$, we have

$$\mathbb{E}\left(W_{(k)}\right) = \frac{k}{q+1}.$$

Once the expected birth and death values are computed, we can use them to plot Betti curves. In Fig 3 example, we generated two random graphs with $p$ = 150 nodes and their edge

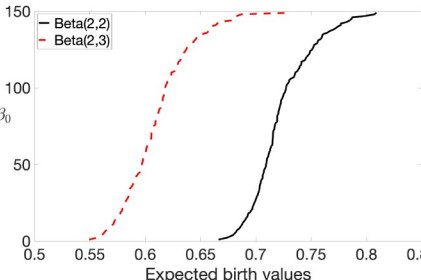 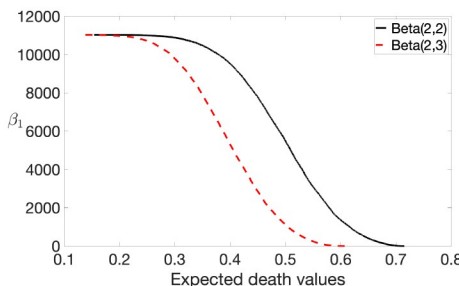

**Fig 3. Plots of Betti-0 (left) and Betti-1 (right) curves of random networks with edges drawn from Beta(2, 2) (in dotted red line) and Beta(2, 3) (in solid black line) distributions.** The change in distribution affects the topology of a network.

weights drawn from Beta(2, 2) and Beta(2, 3) distributions. We observe that a slight change in distribution significantly affects the topology of a network.

## Confidence bands on birth and death values

Given a set of $n$ samples from a random graph $\mathcal{G}(p, W)$, we show how to compute the confidence bands on the expected birth and death values. The method is later validated using a simulation study.

Let the random weights of the $n$ sampled graphs be $w_1, w_2, \ldots, w_n$, where $w_i = (w_{i1}, w_{i2}, \ldots, w_{iq})^\top$ and $w_{ij} \sim F_W$. From the previous section, we know that $W_{(k)}$ follows a asymptotic Gaussian distribution with its mean and variance being

$$\mathbb{E}\left(W_{(k)}\right) \approx F_W^{-1}\left(\frac{k}{q+1}\right) = \mu_k$$

and

$$\mathrm{var}\left(W_{(k)}\right) \approx \frac{\frac{k}{q+1}\left(1 - \frac{k}{q+1}\right)}{(q+1)\left[f_W\left\{F_W^{-1}\left(\frac{k}{q+1}\right)\right\}\right]^2} = \sigma_k^2.$$

The density $f_W$ is estimated by averaging the $n$ graphs with respect to their weights using Gaussian kernel density estimates (KDE). Let the average weight vector be $\bar{w} = \frac{1}{n}\sum_{i=1}^{n} w_i = (\bar{w}_1, \bar{w}_2, \ldots, \bar{w}_q)^\top$. Then, the KDE of the pdf $f_W$ is given by [89]

$$\hat{f}_W(x) = \frac{1}{qh}\sum_{i=1}^{q} K\left(\frac{x - \bar{w}_i}{h}\right),$$

where $K$ is the Gaussian kernel with bandwidth $h$. To estimate $F_W^{-1}$, we first find the empirical cdf of $F_W$ based on the averaged weight vector $\bar{w}$ as

$$\hat{F}_W(x) = \frac{1}{q}\sum_{i=1}^{q} I_{\bar{w}_i \leq x}.$$

Here, $I_A$ is an indicator function takeing value 1 if the event $A$ is true and 0 otherwise. The inverse cumulative distribution of $\hat{F}_W(x)$ is then given by

$$\hat{F}_W^{-1}(x) = \inf\{t \in \mathbb{R} : \hat{F}_W(t) \geq x\}.$$

Once $f_W$ and $F_W^{-1}$ are estimated, we plug-in the corresponding estimates in $\mu_k$ and $\sigma_k^2$ and obtain the estimates $\hat{\mu}_k$ and $\hat{\sigma}_k^2$. Finally, we calculate the $\alpha\%$ confidence intervals as

$$(\hat{\mu}_k - z_\alpha \hat{\sigma}_k, \quad \hat{\mu}_k + z_\alpha \hat{\sigma}_k),$$

where $z_\alpha$ is such that

$$P(Z \geq z_\alpha) = \alpha/2$$

for standard normal $Z \sim \mathcal{N}(0, 1)$. For $\alpha = 95$, we have $z_\alpha = 1.96$.

## Inference on expected birth and death values

Since a graph can be topologically characterized by 0D and 1D barcodes, the topological similarity and dissimilarity between two graphs can be measured using the differences of such barcodes. To quantify these differences, we propose the expected topological loss (ETL) as follows.

Let $\mathcal{G}_1(p, \boldsymbol{U})$ and $\mathcal{G}_2(p, \boldsymbol{V})$ be two random graphs, where the random weights $\boldsymbol{U} = \{U_1, \ldots, U_q\}$ and $\boldsymbol{V} = \{V_1, \ldots, V_q\}$ are drawn from distribution functions $F_U$ and $F_V$, respectively. Further, let the expected birth and death values of $\mathcal{G}_1(p, \boldsymbol{U})$ be

$$F_U^{-1}\left(\frac{i_1}{q+1}\right) < \cdots < F_U^{-1}\left(\frac{i_{m_0}}{q+1}\right) \quad \text{and} \quad F_U^{-1}\left(\frac{j_1}{q+1}\right) < \cdots < F_U^{-1}\left(\frac{j_{m_1}}{q+1}\right).$$

Similarly, let the expected birth and death values of $\mathcal{G}_2(p, \boldsymbol{V})$ be

$$F_V^{-1}\left(\frac{i_1'}{q+1}\right) < \cdots < F_V^{-1}\left(\frac{i_{m_0}'}{q+1}\right) \quad \text{and} \quad F_V^{-1}\left(\frac{j_1'}{q+1}\right) < \cdots < F_V^{-1}\left(\frac{j_{m_1}'}{q+1}\right).$$

Then, the ETL is given by

$$\mathrm{ETL}(\mathcal{G}_1, \mathcal{G}_2) = \sum_{k=1}^{m_0}\left[F_U^{-1}\left(\frac{i_k}{q+1}\right) - F_V^{-1}\left(\frac{i_k'}{q+1}\right)\right]^2 + \sum_{k=1}^{m_1}\left[F_U^{-1}\left(\frac{j_k}{q+1}\right) - F_V^{-1}\left(\frac{j_k'}{q+1}\right)\right]^2. \quad (3)$$

In most applications, the distribution functions $F_U$ and $F_V$ are unknown. In such scenarios, we plug-in the corresponding empirical distribution function estimates $\hat{F}_U^{-1}$ and $\hat{F}_V^{-1}$ in (3). The ETL is a function of expected 0D and 1D barcodes. The expected 0D barcode can also be viewed as the expected heights of branching in a random merge tree [90–95].

## Application of ETL in discriminating networks

The ETL can be used to topologically discriminate between two groups of brain networks. Let $\Omega = \{\Omega_1, \ldots, \Omega_m\}$ and $\Psi = \{\Psi_1, \ldots, \Psi_n\}$ be two sets consisting of $m$ and $n$ complete networks each comprising $p$ number of nodes. Let the empirical distribution functions of the edge weights of the graphs in group $\Omega$ be $\{\hat{F}_{\Omega_1}, \ldots, \hat{F}_{\Omega_m}\}$ and the expected birth and death values for $\Omega_i$ be

$$\hat{F}_{\Omega_i}^{-1}\left(\frac{i_1}{q+1}\right) < \cdots < \hat{F}_{\Omega_i}^{-1}\left(\frac{i_{m_0}}{q+1}\right) \quad \text{and} \quad \hat{F}_{\Omega_i}^{-1}\left(\frac{j_1}{q+1}\right) < \cdots < \hat{F}_{\Omega_i}^{-1}\left(\frac{j_{m_1}}{q+1}\right),$$

which will be simply denoted as

$$u_{1i}^\Omega < \cdots < u_{m_0 i}^\Omega \quad \text{and} \quad v_{1i}^\Omega < \cdots < v_{m_1 i}^\Omega.$$

The average of expected birth and death values within the group $\Omega$ are given by

$$\bar{u}_1^\Omega < \cdots < \bar{u}_{m_0}^\Omega \quad \text{and} \quad \bar{v}_1^\Omega < \cdots < \bar{v}_{m_1}^\Omega.$$

Similarly, for the second group $\boldsymbol{\Psi}$, let the average of expected birth and death values be

$$\bar{u}_1^\Psi < \cdots < \bar{u}_{m_0}^\Psi \quad \text{and} \quad \bar{v}_1^\Psi < \cdots < \bar{v}_{m_1}^\Psi.$$

The test statistic based on ETL to discriminate between groups is then given by

$$\mathcal{L}(\boldsymbol{\Omega}, \boldsymbol{\Psi}) = \sum_{j=1}^{m_0} (\bar{u}_j^\Omega - \bar{u}_j^\Psi)^2 + \sum_{j=1}^{m_1} (\bar{v}_j^\Omega - \bar{v}_j^\Psi)^2. \tag{4}$$

Large $\mathcal{L}(\boldsymbol{\Omega}, \boldsymbol{\Psi})$ indicates a significant topological difference between the two groups whereas a small value suggests that there is no significant topological group difference. Considering the probability distribution of the test statistic $\mathcal{L}(\boldsymbol{\Omega}, \boldsymbol{\Psi})$ is unknown, we use the permutation test [96–99]. In this study, we use 100000 permutations for small scale simulation studies and half million permutations for large-scale real data.

A similar widely-used statistic is the maximum gap statistics. On a similar line to $\mathcal{L}(\boldsymbol{\Omega}, \boldsymbol{\Psi})$, the statistic is given by:

$$\mathcal{L}_1(\boldsymbol{\Omega}, \boldsymbol{\Psi}) = \max_{1 \le j \le m_0} |\bar{u}_j^\Omega - \bar{u}_j^\Psi| + \max_{1 \le j \le m_1} |\bar{v}_j^\Omega - \bar{v}_j^\Psi|. \tag{5}$$

We will use $\mathcal{L}_1(\boldsymbol{\Omega}, \boldsymbol{\Psi})$ to compare with the ETL statistic $\mathcal{L}(\boldsymbol{\Omega}, \boldsymbol{\Psi})$ in the simulation study section.

## Area under Betti curves in discriminating networks

The difference of $\beta_0$ curves can be also quantified using the area under the curve (AUC) [100, 101]. The AUC for $\Omega_i$ of the group $\boldsymbol{\Omega}$ and for $\Psi_j$ of the group $\boldsymbol{\Psi}$ are given by

$$\text{AUC}_{\Omega_i} = \sum_{k=2}^{m_0} k(u_{ki}^\Omega - u_{(k-1)i}^\Omega) \quad \text{and} \quad \text{AUC}_{\Psi_j} = \sum_{k=2}^{m_0} k(u_{kj}^\Psi - u_{(k-1)j}^\Psi).$$

We compute the AUC by summing up the areas of rectangular blocks formed by the expected persistent barcodes. For example, in Fig 1, the area under the Betti-0 curve is $2(W_{(5)} - W_{(3)}) + 3(W_{(6)} - W_{(5)})$.

To determine if AUC is significantly different between groups $\Omega$ and $\Psi$, we use the Wilcoxon rank sum test [102]. The Wilcoxon rank sum test is a nonparametric test of the null hypothesis, For randomly selected values $X$ and $Y$ from two populations, the probability of $X$ being greater than $Y$ is equal to the probability of $Y$ being greater than $X$. This is unlike the previous situation, where we considered a ETL or a max-gap statistic. In those cases, since we are considering the distance between increasing births or deaths of two networks, the consideration of $\mathbb{L}_1$ or $\mathbb{L}_2$ norm in the statistic is meaningful. In contrast to distance-based methods, AUC offers an alternate area based topological inference procedure. The method can be equally applicable for area under Betti-1 curves as well.

## Simulation study

Since there is no ground truth in real brain network data, we performed extensive simulation studies with known ground truth. The Matlab codes for simulation study is provided in https://github.com/laplcebeltrami/orderstat.

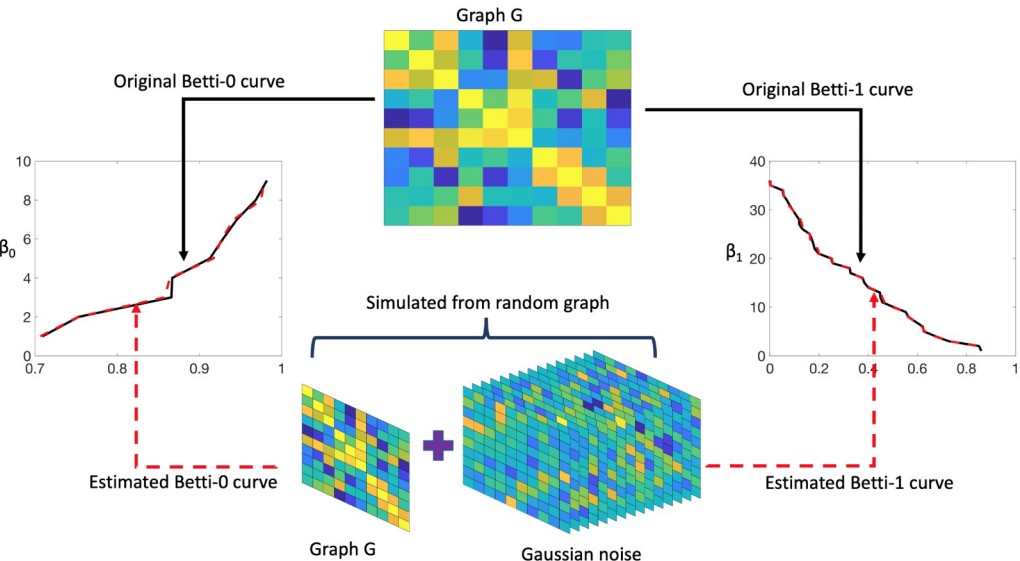

**Fig 4. Schematic of validate the proposed method to compute expected birth and death values.** The ground truth graph *G* is we used to calculate the Betti curves (solid black line) using the birth-death decomposition. We added Gaussian noise to edge weights to generate samples in each group. Then, we apply the proposed method on this set of sampled graphs and estimate the expected birth and death values and the Betti curves (dotted red line).

## Validation of birth and death value estimates

We validate the method to estimate expected birth and death values. We generate the ground truth graph $G(p, \boldsymbol{w})$ with given edge weights and calculate its birth and death values using the birth-death decomposition on $p = 10$ number of nodes [37]. The weight vector $\boldsymbol{w}$ is of dimension $q = p(p-1)/2 = 45$. We drew the $q$ variate random weights $\boldsymbol{w}$ from the Uniform(0, 1) distribution.

We then simulate $n = 15$ vectors of $q$-variate Gaussian noises and add them to the edge weights $\boldsymbol{w}$ of $G(p, \boldsymbol{w})$ to have a set of $n$ graphs

$$G(p, \boldsymbol{w}_i) = G(p, \boldsymbol{w} + \epsilon_i), \quad i = 1, \dots, n, \tag{33}$$

where $\epsilon_i \sim \mathcal{N}_q(\boldsymbol{0}, \sigma^2 \boldsymbol{I})$ with $\sigma = 0.02$. The produced set of graphs $\{G(p, \boldsymbol{w}_1), \dots, G(p, \boldsymbol{w}_n)\}$ is considered as a realizations from a random graph $\mathcal{G}(p, \boldsymbol{W})$ and apply the proposed method to calculate the expected birth and death values along with their corresponding confidence bands. Then, we compare them with the initially calculated birth and death values of $G(p, \boldsymbol{w})$. Fig 4 displays the schematic of the validation procedure. The original and expected birth (left panel) and death (right panel) values are plotted in Fig 5. The black line represents the original birth or death values, the dashed red line indicates the estimated birth or death values, and the dashed blue lines indicate the corresponding 95% upper and lower confidence bands. We observe that the dashed red lines almost overlap the black lines of the original birth and death values. In addition, the original birth or death values almost always lie within the confidence bands supporting the reliability of the proposed methodology.

## Analyzing topological similarity between networks

We provide a toy example to illustrate whether the topological similarity or dissimilarity of two groups of networks, drawn from two different distributions or the same distribution, can

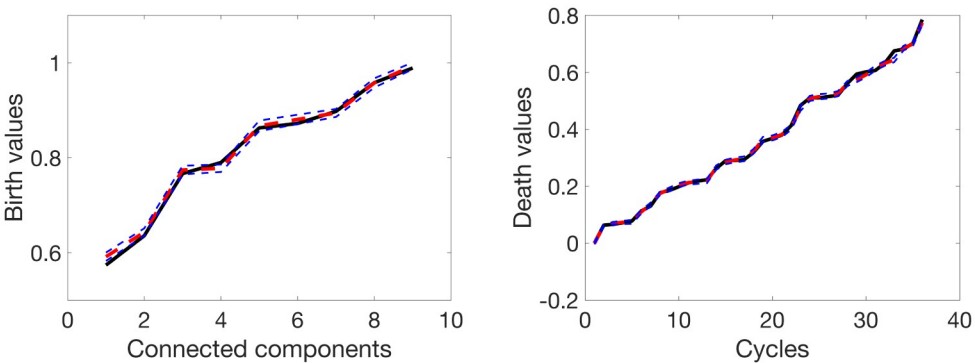

**Fig 5. Plots of the original and expected birth (left) and death (right) values.** The black line represents the original birth (or death) values, the dashed red line indicates the expected birth (or death) values, and the dashed blue lines indicate the corresponding 95% confidence intervals.

be identified using the ETL statistic (4). We used the Beta($a$, $b$) distribution, which are all defined in interval (0, 1). The shape parameters $a$ and $b$ allow for the variety of shapes including the shape of a uniform distribution Uniform(0, 1) when $a = b = 1$. We considered three different distributions (Fig 6).

We generated two groups of networks. For the both groups, we simulated $n = 6, 8, 10$, and 12 networks. We investigated the performance of both small ($p = 10$) and large ($p = 100$)

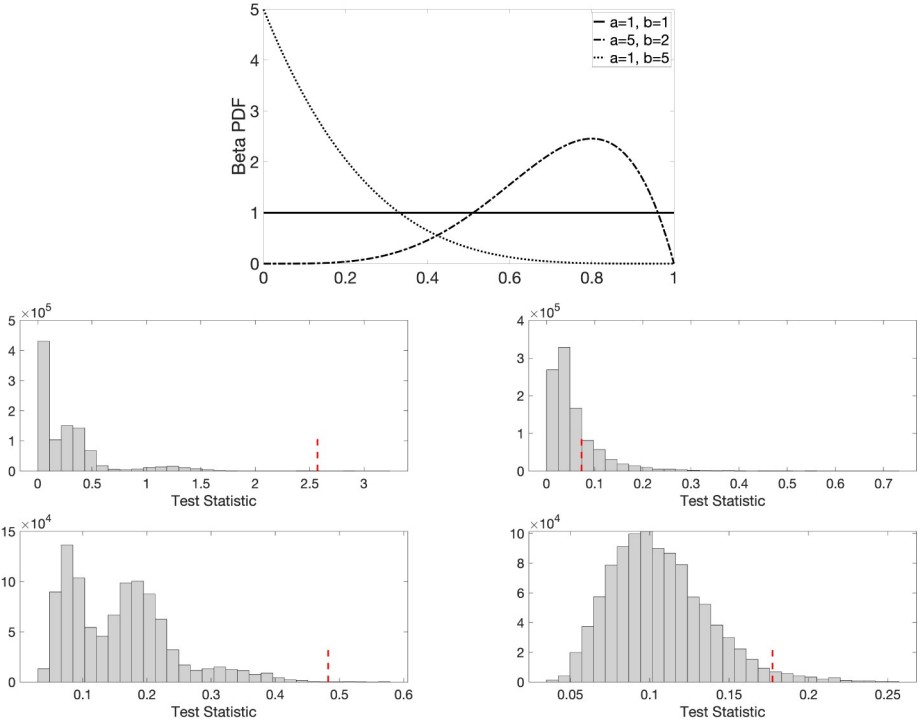

**Fig 6.** Top panel: Probability density functions of Beta(1, 1) or Uniform(0, 1) (in solid line), Beta(5, 2) (in dash-dotted line), and Beta(1, 5) (in dotted line). We sample the edge weights of random graphs from these three different distributions for validation purpose. Middle and bottom panel: Histogram plots of the ETL (middle) and maximum gap (bottom) test statistics and the corresponding observed test statistics (in dotted red lines) for the scenarios: Beta(1, 1) vs. Beta(5, 2) (left) and Beta(1, 1) vs. Beta(1, 1) (right) with 6 networks in each group.

**Table 1. The average _p_-values obtained using the ETL statistic for various pairs of distributions for small (_p_ = 10) and large (_p_ = 100) network settings.** Here, the columns 6 networks, 8 networks, 10 networks, and 12 networks indicate the number of networks that we considered for both the groups. The _p_-values smaller than 0.01 indicate that our method can identify network differences at a 99% confidence level.

| Distribution ($p = 10$) | 6 networks | 8 networks | 10 networks | 12 networks |
|---|---|---|---|---|
| Beta(1, 1) vs. Beta(5, 2) | 0.0017 | $1.00 \times 10^{-5}$ | 0.0000 | 0.0000 |
| Beta(1, 1) vs. Beta(1, 5) | 0.0022 | $1.00 \times 10^{-4}$ | $1.00 \times 10^{-5}$ | 0.0000 |
| Beta(5, 2) vs. Beta(1, 5) | 0.0011 | $1.30 \times 10^{-4}$ | 0.0000 | 0.0000 |
| Beta(1, 1) vs. Beta(1, 1) | 0.2478 | 0.6784 | 0.6859 | 0.8264 |
| Beta(5, 2) vs. Beta(5, 2) | 0.2393 | 0.4497 | 0.7836 | 0.8772 |
| Beta(1, 5) vs. Beta(1, 5) | 0.2721 | 0.6847 | 0.7585 | 0.7573 |
| Distribution ($p = 100$) | 6 networks | 8 networks | 10 networks | 12 networks |
| Beta(1, 1) vs. Beta(5, 2) | 0.0021 | $9.0000 \times 10^{-5}$ | $2.0000 \times 10^{-5}$ | 0.0000 |
| Beta(1, 1) vs. Beta(1, 5) | 0.0018 | $8.0000 \times 10^{-5}$ | $2.0000 \times 10^{-5}$ | $1.0000 \times 10^{-5}$ |
| Beta(5, 2) vs. Beta(1, 5) | 0.0022 | $1.2000 \times 10^{-4}$ | 0.0000 | 0.0000 |
| Beta(1, 1) vs. Beta(1, 1) | 0.6430 | 0.2844 | 0.3308 | 0.2665 |
| Beta(5, 2) vs. Beta(5, 2) | 0.7882 | 0.8828 | 0.5559 | 0.9319 |
| Beta(1, 5) vs. Beta(1, 5) | 0.1526 | 0.3831 | 0.2241 | 0.5021 |

network settings. Small networks may not yield complex cyclic structures often present in large networks. However, the overall conclusions are the same regardless of the size of networks. For the permutation test, we considered 100000 permutations and repeated that 10 times to compute the average _p_-values. Table 1 tabulates the _p_-values for small and large network settings. In all the scenarios, networks drawn from the same distribution produced large _p_-values and networks drawn from different distributions had _p_-values smaller than 0.01. Therefore, we conclude that the proposed ETL statistic, based on expected birth and death values, can discriminate networks drawn from different distributions at 99% confidence level. The middle panel of Fig 6 plots the histograms of the ETL test statistic and the corresponding observed test statistics (in dotted red) for two specific scenarios: (i) Beta(1, 1) vs. Beta(5, 2) (left) and (ii) Beta(1, 1) vs. Beta(1, 1) (right) with 12 networks in each group.

## Comparison of ETL against baselines

We compared the proposed ETL with several other widely-used baseline topological distances such as bottleneck, Gromov-Hausdorff (GH), and Kolmogorov-Smirnov (KS) distances [21, 103, 104]. We also compared the results with the maximum gap statistic defined earlier in (5). In all the scenarios, we considered two groups of networks each of size $n = 6$. The remaining simulation setting is similar to the above. The corresponding _p_-values are presented in Table 2 for small ($p = 10$) and large network ($p = 100$) settings. From the table, we observe that the ETL performs well in most scenarios. In particular, we note that the KS based methods do not perform well whereas the maximum gap based method is quite competitive. Further, for testing no network differences, all the distances perform well.

Since the maximum gap based method exhibits a competitive performance with the ETL based method, we plot the histograms of the maximum gap statistics obtained over different permutations and the corresponding observed test statistics (in dotted red) for two specific scenarios: (i) Beta(1, 1) vs. Beta(5, 2) (left) and (ii) Beta(1, 1) vs. Beta(1, 1) (right) with 6 networks in each group; see the bottom panel of Fig 6. Although both the methods (ETL and maximum-gap) perform well, the ETL generally produces better results (i.e., its _p_-value is closer to 0 when there is a network difference and closer to 1 when there is no network difference).

**Table 2. The average *p*-values obtained using bottleneck, GH, KS, maximum gap, and ETL based statistics for small (*p* = 10) and large (*p* = 100) network settings.** There were 6 networks in each group. The *p*-values smaller than 0.01 indicate that the corresponding method can identify network differences at a 99% confidence level.

| Distribution (*p* = 10) | Bottleneck | GH | KS($\beta_0$) | KS($\beta_1$) | Maximum gap | ETL |
|---|---|---|---|---|---|---|
| Beta(1, 1) vs. Beta(5, 2) | 0.0035 | 0.0028 | 0.4667 | 0.3438 | 0.0022 | 0.0014 |
| Beta(1, 1) vs. Beta(1, 5) | 0.0190 | 0.0692 | 0.3804 | 0.6406 | 0.0016 | 0.0022 |
| Beta(5, 2) vs. Beta(1, 5) | 0.0026 | 0.3345 | 0.2885 | 0.5177 | 0.0021 | 0.0011 |
| Beta(1, 1) vs. Beta(1, 1) | 0.2255 | 0.4385 | 0.2591 | 0.6893 | 0.1013 | 0.3136 |
| Beta(5, 2) vs. Beta(5, 2) | 0.3046 | 0.2346 | 0.1991 | 0.6035 | 0.1446 | 0.2393 |
| Beta(1, 5) vs. Beta(1, 5) | 0.3351 | 0.5392 | 0.1217 | 0.4172 | 0.1058 | 0.2721 |
| Distribution (*p* = 100) | Bottleneck | GH | KS($\beta_0$) | KS($\beta_1$) | Maximum gap | ETL |
| Beta(1, 1) vs. Beta(5, 2) | 0.0029 | 0.0032 | 0.0879 | 0.2984 | 0.0020 | 0.0021 |
| Beta(1, 1) vs. Beta(1, 5) | 0.0039 | 0.1391 | 0.3412 | 0.1333 | 0.0032 | 0.0028 |
| Beta(5, 2) vs. Beta(1, 5) | 0.0001 | 0.8790 | 0.5854 | 0.4600 | 0.0015 | 0.0022 |
| Beta(1, 1) vs. Beta(1, 1) | 0.8204 | 0.3911 | 0.9848 | 0.7357 | 0.0863 | 0.6430 |
| Beta(5, 2) vs. Beta(5, 2) | 0.5272 | 0.0919 | 0.7677 | 0.6115 | 0.3785 | 0.7882 |
| Beta(1, 5) vs. Beta(1, 5) | 0.4840 | 0.1640 | 0.4224 | 0.6654 | 0.1315 | 0.1526 |

## Area under Betti curves

We also conducted a simulation study for the method based on the area under $\beta_0$ curve. The considered simulation layout was the same as before. The obtained *p*-values are tabulated in Table 3 for small networks (*p* = 10) and large networks (*p* = 100). As shown in Fig 3, a slight change in distribution significantly changes the topology of the network and, therefore, the area under $\beta_0$ curve varies significantly. This change is more visible especially for large networks, which incases AUC. However, the Wilcoxon rank sum test places ranks to the aggregated sample that combined the first and second sample, then considers the sum of ranks for the both samples. This makes the test statistic fairly robust even if the distributions are varied. This makes the *p*-value computation extremely stable for large networks. For example Table 3 shows the exactly same *p*-value of 0.0022 for *p* = 100. Similar to the ETL, this approach can also discriminate networks drawn from different distributions at a 99% confidence level.

**Table 3. The average *p*-values obtained using Wilcoxon rank sum test on the areas under $\beta_0$ curves for small (*p* = 10) and large (*p* = 100) network settings.** Here, the columns 6 networks, 8 networks, 10 networks, and 12 networks indicate the number of networks that we considered for both the groups. The *p*-values smaller than 0.01 indicate that our method can identify network differences at a 99% confidence level.

| Distribution (*p* = 10) | 6 networks | 8 networks | 10 networks | 12 networks |
|---|---|---|---|---|
| Beta(1, 1) vs. Beta(5, 2) | 0.0087 | $3.10 \times 10^{-4}$ | $7.68 \times 10^{-4}$ | 0.0014 |
| Beta(1, 1) vs. Beta(1, 5) | 0.0022 | $6.21 \times 10^{-4}$ | 0.0017 | 0.0061 |
| Beta(5, 2) vs. Beta(1, 5) | 0.0043 | 0.0030 | $1.82 \times 10^{-4}$ | $3.65 \times 10^{-5}$ |
| Beta(1, 1) vs. Beta(1, 1) | 0.6991 | 0.7209 | 0.4727 | 0.8852 |
| Beta(5, 2) vs. Beta(5, 2) | 0.4848 | 0.2786 | 0.2413 | 0.9770 |
| Beta(1, 5) vs. Beta(1, 5) | 0.8182 | 0.2786 | 0.7337 | 0.5444 |
| Distribution (*p* = 100) | 6 networks | 8 networks | 10 networks | 12 networks |
| Beta(1, 1) vs. Beta(5, 2) | 0.0022 | $1.5540 \times 10^{-4}$ | $1.8365 \times 10^{-4}$ | $3.6585 \times 10^{-5}$ |
| Beta(1, 1) vs. Beta(1, 5) | 0.0022 | $1.5540 \times 10^{-4}$ | $1.8267 \times 10^{-4}$ | $3.6585 \times 10^{-5}$ |
| Beta(5, 2) vs. Beta(1, 5) | 0.0022 | $1.5540 \times 10^{-4}$ | $1.8267 \times 10^{-4}$ | $3.6585 \times 10^{-5}$ |
| Beta(1, 1) vs. Beta(1, 1) | 0.9372 | 0.2345 | 0.2413 | 0.4025 |
| Beta(5, 2) vs. Beta(5, 2) | 0.8182 | 0.1605 | 0.6776 | 0.7075 |
| Beta(1, 5) vs. Beta(1, 5) | 0.2403 | 0.6454 | 0.7913 | 0.1572 |

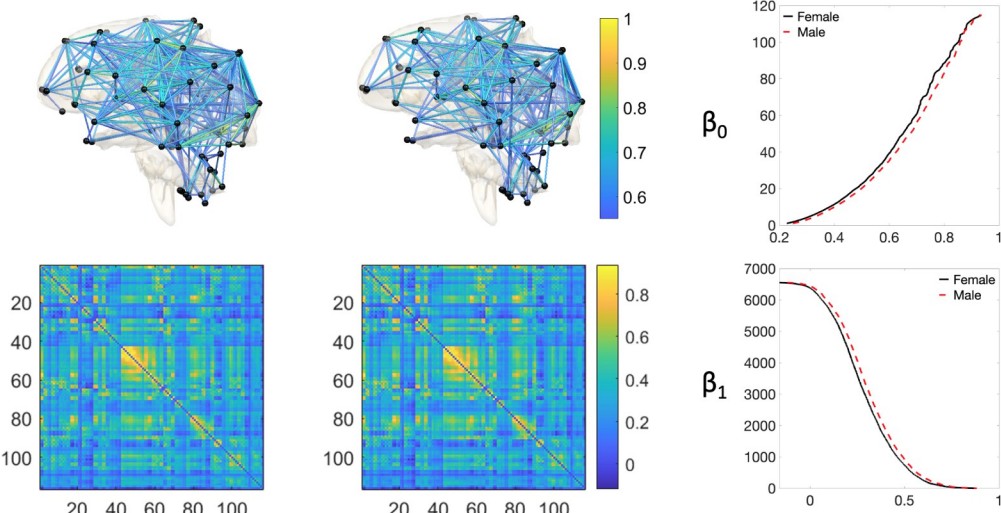

**Fig 7. Visualization of the fMRI brain data.** Left panel: The average female brain network (top) and the corresponding correlation matrix (bottom). Middle panel: The average male brain network (top) and the corresponding correlation matrix (bottom). Right panel: The $\beta_0$ (top) and $\beta_1$ (bottom) curves for female (in solid black) and male (in dashed red) brain networks. For a better visualization, we consider a threshold value of 0:5 while plotting the brain networks so that they contain fewer number of edges.

## Results

For each of the 400 subjects, we computed the whole-brain functional connectivity by calculating the Pearson correlation matrix over 1200 time points across 116 anatomical regions resulting in 400 correlation matrices of dimension $116 \times 116$. Therefore, using our notations, we have $p = 116$ nodes, $q = p(p-1)/2 = 6670$ edges, $m_0 = p - 1 = 115$, and $m_1 = (p-1)(p-2)/2 = 6555$. Fig 7 summarizes the average female (left) and male (right) brain networks (top) and the corresponding correlation matrices (bottom).

### Two-sample test using ETL statistic

Given the 400 correlation matrices of 168 males and 232 females, we aim to check whether the proposed ETL statistic can determine the difference between the groups of males and females. We assume that the male and female edge weights are coming from distributions with cdfs $F_U$ and $F_V$, respectively. However, these distribution functions are unknown. Therefore, we need to estimate them because the ETL statistic is constructed using these cdfs. To estimate the cdf, we average the male (female) correlation matrices across 168 subjects (232 subjects) and find the empirical cdf based on the averaged 6670 edge weights. The empirical cdfs of the average edge weights of females (in solid black line) and males (in dashed red line) are presented in the left panel of Fig 8. We observe that the empirical cdf corresponding to female is slightly higher than that of male. This suggests a relatively more number of edge weights with smaller values for the female, and a relatively more number of edge weights with bigger values for the male. In other words, the distribution of the female edge weights is slightly positively skewed than the male edge weights. Fig 9 plots the $\beta_0$ and $\beta_1$ curves of the average female and male networks (calculated in the standard way) and their corresponding estimated counterparts (computed using the expected birth and death values). We observe that the estimated Betti curves well approximate the original Betti curves.

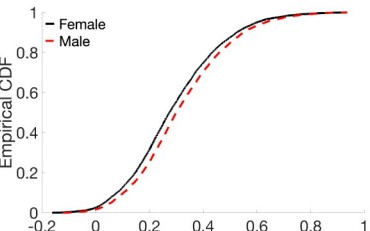
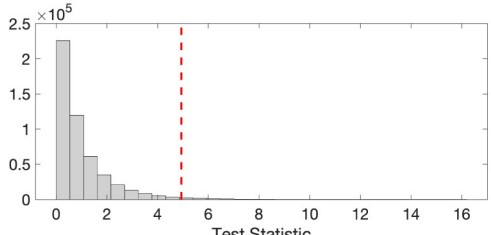

**Fig 8.** Left: Plot of the empirical cdfs of the average edge weights of females (in solid black line) and males (in dashed red line). Right: Histogram plot of the ETL statistic based on the resting-state fMRI dataset. The dotted red line represents the observed value of the ETL statistic.

To conduct the test, we used the permutation test with 500000 random permutations. The observed test statistic is 4.9372 and the *p*-value is 0.0134. The histogram of test statistic is plotted in the right panel of Fig 8. We conclude that, although the weight distributions of the males and females are very close, the proposed ETL statistic can still discriminate them at a 95% confidence level.

## Two-sample test using AUC statistic

We conducted a two-sample test using the method based on the area under $\beta_0$ curve. The observed value of the Wilcoxon rank-sum statistic is 48374. The statistic corresponds to the *p*-value of 0.1036. That is, the test fails to discriminate male and female subjects if we use the

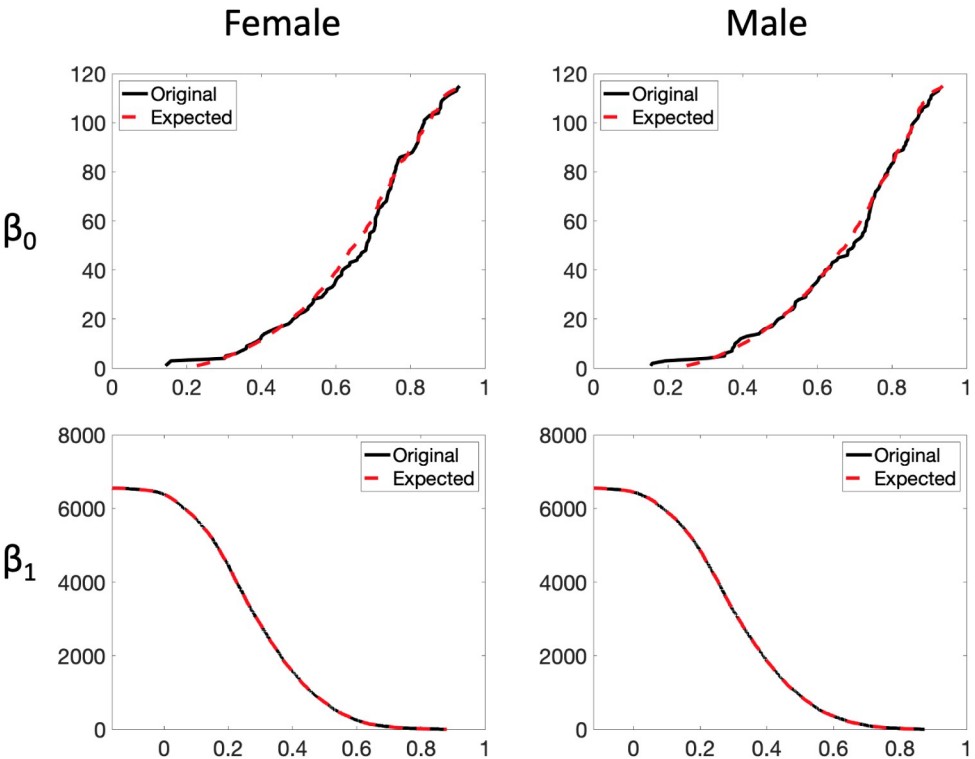

**Fig 9.** Plots of the original (solid black line) and estimated (dashed red line) $\beta_0$ (top) and $\beta_1$ (bottom) curves using expected birth and death values for the female (left) and male (right) brain networks.

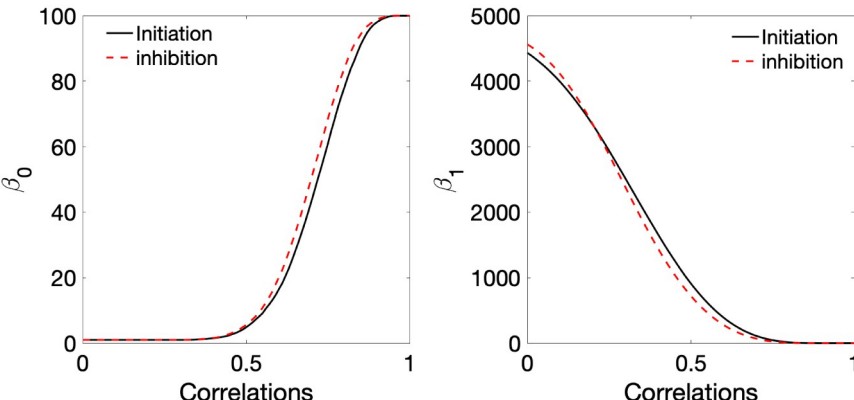

**Fig 10. The average Betti curves of obtained from the graph filtration on correlation matrices computed separately for the inhibition (go) and initiation (no-go) blocks of fMRI time series in cognitive aging study [130, 131].**

traditional values of $\alpha$, the level of significance, to be 0.05 or 0.1. However, if we relax this assumption a bit, the test can discriminate males and females at a confidence level of 89.5%.

The sex differences of resting state functional networks were previously investigated. There is known sex difference in the parietal region involved in spatial ability [105]. [106] reported sex differences in the left parietal, precentral and postcentral regions. The sex difference is also reported in the left rolandic operculum [107]. The previous rs-fMRI studies mainly focused on brain region specific analysis and not topological. Our topological methods are different. The use of the order statistic in quantifying topological difference between males and females is novel. This method identifies the impact of distribution differences in topological features. These specific results have not been observed before to best of our knowledge.

## Discussion

The concept of random graphs was first proposed in mid-twentieth century [39] and has been of many researchers' interest ever since [108–112]. The concepts of TDA tools such as persistent barcodes were extended to handle stochastic cases, which triggered the computation of expected persistent barcodes. However, such computation may require complex theoretical constructs. In this article, we considered a random graph model for which the computation of expected persistent barcodes became simplified by using the *order statistic*.

[37] formulated a topological loss based on the birth and death values of connected components and cycles of a network that provided an optimal matching and alignment at the edge level. In this article, we extended this formulation to a random graph scenario and proposed the expected topological loss (ETL) based on the expected birth and death values. We use the ETL as a test statistic to discriminate between two groups of networks. We validated this method using a simulation study. We showed that the ETL can identify group differences at a 99% confidence level whereas it produces large *p*-values when there is no network differences. We compared the proposed approach with baseline approaches and established an overall superior performance of the proposed method. Further, we considered the area under the Betti curves [101]. This resulted a scalar quantification of the curves which was used to discriminate between the groups. A respective simulation study showed its successful discriminative ability whenever there are network differences. We also applied the developed tools in a resting-state brain fMRI dataset and showed that they can differentiate male and female brain networks.

To calculate the expected persistent barcodes, we computed the unknown distribution using the nonparametric empirical distribution function. However, one may also consider hierarchical or Bayesian parametric models for the edge weights instead. For example, one may consider the edge weights to be drawn from a $\mathcal{N}(\mu, \sigma^2)$ distribution, where the location parameter $\mu$ and the dispersion parameter $\sigma^2$ have a Gaussian and an inverse gamma conjugate prior, respectively. The parameters of the prior distributions will allow flexibility while we can still enjoy the advantages of a parametric model. This direction can be pursued in the future.

We can also use different filtration schemes such as relative filtration [113] or normalized filtration that scales filtration values between 0 and 1 [114]. As long as the order of sorted edge weights are not changed, they will not affect the statistical results. The Wasserstein distance we used is defined on the sorted edge weights. As long as we do not change the value of edge weights, the statistical results will not change.

Our methodology is based on the graph filtration, which gives both 0D and 1D persistence as monotonic 1D functions of birth and death values only. On the other hand, the clique filtration [115], does not produce monotone persistence or barcodes and the proposed method is not directly applicable [116–121]. Our method is applicable to any filtration that provides monotone persistence or Betti curves. The proposed graph filtration computes the barcodes in $\mathcal{O}(p \log p)$, which is significantly faster than Rips filtrations. In traditional Rips filtrations, the computational complexity grows rapidly with the number of simplices [122]. With $p$ nodes, the size of the k-skeleton grows as $p^{k+1}$ and the computational run time is $\mathcal{O}(p^{3k+3})$ [123, 124]. Compared to the graph filtration, the Rips filtration constructed using Ripser package [125] is about 8 times slower in a computer. On top of that, we also need to compute the Wasserstein distance between persistent diagrams [126]. The Wasserstein distance computation requires expensive optimization process involving $\mathcal{O}(p^6)$ run-time [127, 128].

Our algorithm exploits the geometric structure of the graph filtration, resulting in the persistence diagram representation in the form of 1-dimensional sorted scalar values. Thus, the proposed method computes he Wasserstein distance in $\mathcal{O}(p \log p)$ bypassing multitude of computational bottlenecks. For resampling based statistical inference such as the permutation test, the computational bottleneck is caused by repreatedlby computing the test statistic over the random permutations of group labels at least half million times [129]. This is impractical if the Wasserstein distance for the Rips filtrations has to be used for 400 networks and then the whole computation has to be done repeatedly half million times. The development of scalable computation will have a significant impact in resampling based statistical inference.

Graph filtrations produce the monotone birth and death values over filtrations. Since our birth and death values are exactly edge weights, the slight changes in edge weight distribution will correspondingly change the birth and death values slightly. Since the expected topological loss (ETL) is sum of differences of birth and death values, it will also change correspondingly. This is the very reason which assures that our method can successfully discriminate topological group differences whenever there is a difference in edge distributions between two groups.

The monotone Betti curves usually follow S-shaped curves similar to the pattern of cumulative distribution functions. The pattern of S-shaped Betti curves do not change drastically even if different dataset is used as long as they are weighted graphs. To demonstrate this, we plotted Betti curves for task-fMRI. Fig 10 shows the Betti curves for task-fMRI, where cognitive inhibition was measured using go/no-go paradigm in 144 subjects on 100 brain regions [130, 131]. The correlation matrices were computed separately for the inhibition (go) and initiation (no-go) blocks of fMRI time series. The monotonic Betti curves show almost identical pattern of Betti curves in rs-fMRI networks in our study.

## Acknowledgments

We like to thank Shih-Gu Huang of the National University of Singapore for providing support for fMRI processing. We also thank Gary Shiu of University of Wisconsin-Madison for discussion on the computational runtime on Rips filtrations.

## Author Contributions

**Conceptualization:** Moo K. Chung.

**Formal analysis:** Soumya Das.

**Funding acquisition:** Moo K. Chung.

**Methodology:** Soumya Das.

**Project administration:** Moo K. Chung.

**Resources:** Moo K. Chung.

**Supervision:** Moo K. Chung.

**Validation:** Soumya Das.

**Visualization:** D. Vijay Anand.

**Writing – original draft:** Soumya Das, D. Vijay Anand.

**Writing – review & editing:** Moo K. Chung.

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
