## [Decision Letter · Decision Letter 0]

20 Jun 2022

PONE-D-22-10228Topological Data Analysis of Brain Networks Through Order StatisticsPLOS ONE

Dear Dr. Chung,

Thank you for submitting your manuscript to PLOS ONE. After careful consideration, we feel that it has merit but does not fully meet PLOS ONE’s publication criteria as it currently stands. Therefore, we invite you to submit a revised version of the manuscript that addresses the points raised during the review process.

 The first reviewer has serious concerns about the manuscript, and in particular, has stated that the approach is essentially a renamed version of an existing approach (hierarchical clustering). In order to move your manuscript forward, I'd ask you to please address this point directly (both in the paper itself and in a revision letter) either by pointing out how your approach is not identical, thereby refuting the critique, or by modifying your approach accordingly. The second reviewer is more satisfied with the manuscript, but has suggested a number of improvements that you should address.

We look forward to receiving your revised manuscript.

Kind regards,

Chad M. Topaz

Academic Editor

PLOS ONE

Journal Requirements:

“NO”

5. Please expand the acronym “NIH, NSF and NIH” (as indicated in your financial disclosure) so that it states the name of your funders in full.

6. Thank you for stating the following in your Competing Interests section: 

“NO”

7. Thank you for stating the following in the Acknowledgments Section of your manuscript:

“We like to thank Shih-Gu Huang of National University of Singapore for providing support for fMRI processing. This study is funded by NIH R01 EB02875 and NSF MDS-2010778.”

“NO”

8. Please amend either the title on the online submission form (via Edit Submission) or the title in the manuscript so that they are identical.

9. Please ensure that you refer to Figure 7 in your text as, if accepted, production will need this reference to link the reader to the figure.

Reviewers' comments:

Reviewer's Responses to Questions

**Comments to the Author**

1. Is the manuscript technically sound, and do the data support the conclusions?

Reviewer #1: Partly

Reviewer #2: Yes

2. Has the statistical analysis been performed appropriately and rigorously? 

Reviewer #1: I Don't Know

Reviewer #2: Yes

3. Have the authors made all data underlying the findings in their manuscript fully available?

Reviewer #1: Yes

Reviewer #2: Yes

4. Is the manuscript presented in an intelligible fashion and written in standard English?

Reviewer #1: Yes

Reviewer #2: Yes

5. Review Comments to the Author

Reviewer #1: In this article, the authors describe an order-statistic based approach to classification of brain networks.

1. Is the manuscript technically sound, and do the data support the conclusions?

I cannot recommend this article for publication based on the technical methods used. The analysis is described as an application of topological data analysis (TDA), but the actual tools used are more correctly described in terms of classical hierarchical clustering. Specifically, the authors develop their methods only in the context of the 1D graph, not any enrichment into a simplicial complex. Since they can study only 0D and 1D homology in this setting, and they discard the barcode in favor of lists of birth/death times, it turns out that they are using an impoverished version of hierarchical single linkage clustering, as I will make clear below.

Explicitly, they rely on the classical graph theory fact that, if one removes the (n choose 2) edges of a complete graph in some order, at exactly (n-1) steps, the number of components will increase. Thus, the rest will decrease the number of circuits. As the authors track only the locations of these changes -- the "birth" and "death" times of bars in the 0D and 1D barcode in this very simple case -- as they note, the steps which increase the number of components are the complement of those that decrease the number of circuits. However, this tells us that the data contained in the list of 1D death times is entirely determined by the list of 0D birth times; we simply take the complement in the ordered list (1..(n choose 2)). Thus, the 1D Betti curve they describe contains no information not available using the 0D Betti curve and the size N of the network. The information they retain is the data of the heights of splitting of components in the merge tree, but not the data of which branches are attached to which. As such, the statistic being used in the paper is effectively the expected heights of branching in a random merge tree. I am not sure if this is well-studied in the literature, but it seems like a careful review of the relevant work -- some of it stretching back decades -- should be performed before the authors can claim novelty, and in any case that prior work by the community that studies such objects should be cited. Further, the authors should be careful to cite primary sources for TDA methods; many of the ideas they discuss here first appear in early work of e.g. Kahle on homology of random graphs, and various groups' early work on statistical TDA for data analysis.

In addition, the authors claim that their statistic is useful for reducing the computational burden of topological data analysis. However, on networks of the size the authors are studying, modern software will easily compute persistence diagrams up through degrees three or four on a standard laptop, and much larger networks can be studied on even moderately powerful servers. Thus, the examples in the paper do not serve to demonstrate utility of the method.

Finally, the authors repeatedly claim that this information completely characterizes the "topology" of a graph. If by this they mean that the branching heights of a merge tree are a complete invariant of weighted complete graphs, we can see that this is untrue simply by noting that there are multiple non-isomorphic merge trees with the same list of branch heights: consider a tree with two branches where both attach to the trunk versus one where the shorter branch attaches to the longer.

2. Has the statistical analysis been performed appropriately and rigorously?

The controls and statistical tests appear correctly done to me, but given that the authors do not survey or discuss the literature about statistics on merge trees, it is not clear to me that more appropriate analyses should not have been performed.

4. Is the manuscript presented in an intelligible fashion and written in standard English?

There are several places in the manuscript where careful editing would improve clarity. In particular, the subscripted indices i_a and j_a used in the definition of the order statistic are used for pages without addressing how one would go about finding them, or even acknowledging that the authors have a method for doing so. In addition, the above-mentioned claim about characterizing topology is unclear without further explanation, as are a range of claims made about TDA.

Reviewer #2: Review: Topological Data Analysis of Brain Networks Through Order Statistics

The authors develop order statistics to compute expected Betti curves for the graph filtration of weighted networks. Specifically, they propose the expected topological loss (ETL) as a test statistic. They showcase the ETL on random graphs with 10 nodes and compare it to the maximum gap statistics, the area under the Betti-0 curve, the bottleneck distance, the Gromov-Hausdorff distance, and the Kolmogorov-Smirnov distance. They then apply the ETL to weighted networks with 116 nodes created from resting state fMRI data. Using the approach on the data, they find a statistically significant topological difference between male and female brain networks.

The manuscript is well written and addresses an interesting question. We recommend that the manuscript be accepted with minor revisions (see comments below):

p4: It would be helpful to give the reader an intuition about what you mean by independent cycles. Similarly, a brief explanation how to compute the total number of cycles would be helpful.

p4: Since you mention that you consider the edge weights as the filtration values rather than, for example, relative filtration steps or a normalised filtration - how would such a change in filtration affect your developed statistics?

p4: You only consider edges and nodes in your filtration giving only barcodes in H0 and H1. Could your approach be generalised to a clique filtration, which has in the past also been applied in neuroscience contexts?

p8: Since slight changes in weight distribution significantly change the topology of the network, is the distribution of edge weights in the data comparable enough between different subjects to motivate group level summaries? Does this aspect change when moving from resting state to task-based fMRI data?

p7: Figure 2 is referred to very late in the text. An earlier reference would be helpful for the reader.

p14: What rationale is behind choosing networks that are very small compared to the data? I.e. 10 nodes versus 116?

p17: Has this difference in male and female brain networks been observed in this type of data before?

6. PLOS authors have the option to publish the peer review history of their article (what does this mean?). If published, this will include your full peer review and any attached files.

Reviewer #1: No

Reviewer #2: No

---

## [Author Response · Author response to Decision Letter 0]

1 Sep 2022

We have attached our response letter in PDF.

---

## [Decision Letter · Decision Letter 1]

7 Oct 2022

Topological Data Analysis of Brain Networks Through Order Statistics

PONE-D-22-10228R1

Dear Dr. Chung,

We’re pleased to inform you that your manuscript has been judged scientifically suitable for publication and will be formally accepted for publication once it meets all outstanding technical requirements.

Additionally, please note this one small, remaining comment from one of the reviewers. I would ask you to please address this in your finalized manuscript:

Regarding the clarification of independent cycles on p.5 I think there is an error: the complete graph on four vertices should have three independent cycles rather than two (how would you obtain the cycle [W_1]+[W_3]+[W_6] from the two independent cycles given in the example?). Also, it wasn’t entirely clear to me why both G_0 and G_{-\\inf} were used to refer to the complete graph (neither notation was used in Fig. 1). The example would further be much clearer if the complete graph with labelled edges were included in the figure.

Within one week, you’ll receive an e-mail detailing the other required amendments. When these have been addressed, you’ll receive a formal acceptance letter and your manuscript will be scheduled for publication.

Kind regards,

Chad M. Topaz

Academic Editor

PLOS ONE

Additional Editor Comments (optional):

Reviewers' comments:

Reviewer's Responses to Questions

**Comments to the Author**

1. If the authors have adequately addressed your comments raised in a previous round of review and you feel that this manuscript is now acceptable for publication, you may indicate that here to bypass the “Comments to the Author” section, enter your conflict of interest statement in the “Confidential to Editor” section, and submit your "Accept" recommendation.

Reviewer #1: All comments have been addressed

Reviewer #2: (No Response)

2. Is the manuscript technically sound, and do the data support the conclusions?

Reviewer #1: Yes

Reviewer #2: Yes

3. Has the statistical analysis been performed appropriately and rigorously? 

Reviewer #1: Yes

Reviewer #2: Yes

4. Have the authors made all data underlying the findings in their manuscript fully available?

Reviewer #1: Yes

Reviewer #2: Yes

5. Is the manuscript presented in an intelligible fashion and written in standard English?

Reviewer #1: Yes

Reviewer #2: Yes

6. Review Comments to the Author

Reviewer #1: (No Response)

Reviewer #2: Thank you for addressing my comments. Overall, I am happy with the improvements to the manuscript. Regarding the clarification of independent cycles on p.5 I think there is an error: the complete graph on four vertices should have three independent cycles rather than two (how would you obtain the cycle [W_1]+[W_3]+[W_6] from the two independent cycles given in the example?). Also, it wasn’t entirely clear to me why both G_0 and G_{-\\inf} were used to refer to the complete graph (neither notation was used in Fig. 1). The example would further be much clearer if the complete graph with labelled edges were included in the figure.

7. PLOS authors have the option to publish the peer review history of their article (what does this mean?). If published, this will include your full peer review and any attached files.

Reviewer #1: No

Reviewer #2: No

---

## [Editor Report · Acceptance letter]

25 Oct 2022

PONE-D-22-10228R1 

Topological Data Analysis of Human Brain Networks Through Order Statistics 

Dear Dr. Chung:

I'm pleased to inform you that your manuscript has been deemed suitable for publication in PLOS ONE. Congratulations! Your manuscript is now with our production department. 

Kind regards, 

on behalf of

Dr. Chad M. Topaz 

Academic Editor

PLOS ONE